# How Much Can We Forget about Data Contamination?

**Sebastian Bordt** [1]   **Suraj Srinivas** [2]   **Valentyn Boreiko** [1]   **Ulrike von Luxburg** [1]

## Abstract

The leakage of benchmark data into the training data has emerged as a significant challenge for evaluating the capabilities of large language models (LLMs). In this work, we challenge the common assumption that small-scale contamination renders benchmark evaluations invalid. First, we experimentally quantify the magnitude of benchmark overfitting based on scaling along three dimensions: The number of model parameters (up to 1.6B), the number of times an example is seen (up to 144), and the number of training tokens (up to 40B). If model and data follow the Chinchilla scaling laws, minor contamination indeed leads to overfitting. At the same time, even 144 times of contamination can be forgotten if the training data is scaled beyond five times Chinchilla, a regime characteristic of many modern LLMs. Continual pre-training of OLMo-7B corroborates these results. Next, we study the impact of the weight decay parameter on example forgetting, showing that empirical forgetting occurs faster than the cumulative weight decay. This allows us to gauge the degree of example forgetting in large-scale training runs, indicating that many LLMs, including Llama 3 405B, have forgotten the data seen at the beginning of training.

## 1. Introduction

A core principle of machine learning is that a model should not be trained on the test set used for evaluation (Donoho, 2017). For foundation models trained on Internet-scale data, there are increasing concerns that this principle is violated due to the leakage of benchmark evaluation data into the training data (Xu et al., 2024; Oren et al., 2024). Indeed, many LLM developers have found overlap between

their training data and the benchmark questions used for evaluation (Brown et al., 2020; Dubey et al., 2024). What is more, research on memorization (Carlini et al., 2019; 2021) shows that text sequences from the training data are sometimes encoded within the model, including machine learning datasets (Grynbaum & Mac, 2023; Liang et al., 2023; Nasr et al., 2023; Bordt et al., 2024).

While the fact that data contamination *can* lead to invalid performance evaluations is now well-established (Magar & Schwartz, 2022; Li & Flanigan, 2024; Yang et al., 2023; Jiang et al., 2024), little is known about the precise conditions under which this is the case. The main reason for this is that the training data is usually unknown, and contamination identified via clever research designs that work around this restriction (Golchin & Surdeanu, 2023; Oren et al., 2024; Deng et al., 2024). Because modern foundation models are sometimes trained for over a million gradient steps (Dubey et al., 2024), it is unclear whether a single update on contaminated data at some point during training necessarily impacts downstream evaluations. And indeed, there is quite some evidence that language models need to see samples repeatedly to have any impact on the final model. For example, many papers on memorization have found that it occurs only when a sample is frequently repeated in the training data (Carlini et al., 2022; Biderman et al., 2023; Huang et al., 2024b). The same is true for research on knowledge acquisition, where a fact needs to be paraphrased many times before it is finally remembered by the model (Allen-Zhu & Li, 2023; Cao et al., 2024; Chang et al., 2024).

In this work, we study the impact of data contamination in a controlled setting.[1] This means we train language models from scratch on datasets where we explicitly insert contaminated examples (Jiang et al., 2024). We begin by quantifying how the overall magnitude of benchmark overfitting (or the cross-entropy loss of an observed sample) changes as we **scale along three critical dimensions**: (1) the number of model parameters, (2) the number of training tokens, and (3) the number of repetitions of an example in the training data (Section 4.1). Holding the other two dimensions fixed, we find that *the effect of scaling is monotone in each dimension.*

---

[1]University of Tübingen, Tübingen AI Center, Germany [2]Bosch Research North America & Bosch Center for Artificial Intelligence (BCAI), Sunnyvale, USA. Correspondence to: Sebastian Bordt <sebastian.bordt@uni-tuebingen.de>.

*Proceedings of the $42^{nd}$ International Conference on Machine Learning*, Vancouver, Canada. PMLR 267, 2025. Copyright 2025 by the author(s).

[1]Code is available at https://github.com/tml-tuebingen/forgetting-contamination/.

First, similar to many other works, we find that the tendency of a model to overfit increases in the number of parameters (Goodfellow et al., 2016; Zhang et al., 2017; Carlini et al., 2022). Second, and this is also expected, we find a clear scaling in the number of repetitions, where more frequently repeated observations exhibit stronger overfitting (Carlini et al., 2022; Huang et al., 2024b). More surprisingly, we find that the effect of contamination can *vanish* as we increase the number of training tokens, up to the point where 12 repetitions of an entire dataset in the training data have no impact on the downstream evaluation *on that same dataset*.

Our investigation reveals that the **forgetting** dynamics of gradient descent (Tirumala et al., 2022; Jagielski et al., 2023) is the reason why increasing the number of tokens alleviates the impact of contamination. Concretely, we show that training on five times Chinchilla (Hoffmann et al., 2022) of clean data can cause a model to forget even 144 times repeated training examples (Section 4.2). Forgetting also occurs for larger models, a point that we demonstrate using OLMo-7B (Groeneveld et al., 2024) (Section 4.3). What is the reason for the forgetting? We show that exposure to novel data is important. Interestingly, models tend to exhibit the strongest overfitting on examples seen repeatedly throughout training, even compared to those seen during the end (Section 4.2).

Because running pre-training experiments is expensive, we also ask to what degree forgetting can be explained by the **training dynamics of gradient descent**. We find that the weight decay parameter and learning rate schedule of the AdamW optimizer (Loshchilov & Hutter, 2019) play a key part in forgetting past training examples (Section 5). Concretely, we demonstrate that the cumulative weight decay (Section 5.1) bounds empirical forgetting (Section 5.2). This approach allows us to gauge the degree of forgetting in large-scale training runs by analyzing the optimization hyperparameters (Section 5.3). It also allows us to approximate how the final model parameters balance the gradient updates from different stages of training. Our analysis indicates that many LLMs, including OLMo-7B and Llama 3 405B (Dubey et al., 2024), have forgotten the data seen at the beginning of training.

Taken together, our **main contribution** is to show that the impact of individual examples in the training data depends on the precise characteristics of the setting. There are settings where the effect can be significant; Chinchilla training is an important example (Section 4.1). However, there are equally realistic settings where individual examples don't matter - including quite likely the data-intensive training runs of many recent LLMs (Gemma Team, 2024).

Supplement Sections A.1 and A.2 provide an overview of the most important takeaways and limitations of our work.

## 2. Related Work

**Data Contamination.** The GPT-3 paper (Brown et al., 2020) uses an n-gram-based approach to differentiate between "clean" and "dirty" benchmark questions. This approach has since been used in many LLM reports (Chowdhery et al., 2023; Touvron et al., 2023), including Llama 3 (Dubey et al., 2024), where it is estimated that there might be a performance gain of up to 8 and 14 percentage points on PiQA and HellaSwag, respectively. The GPT-4 technical report (Achiam et al., 2023) remarkably concluded that *"contamination overall has very little effect on the reported results"*. This has since given rise to a literature that aims to *detect* (Oren et al., 2024), *mitigate* (Li et al., 2024), and *estimate the effect of* (Yang et al., 2023; Bordt et al., 2024) data contamination under various assumptions, but crucially without access to the training data. This literature often challenges the conclusion that contamination overall has little effect in GPT-4 (Xu et al., 2024). In closely related work, Jiang et al. (2024) show that inserting benchmark questions into the pre-training data can lead to overfitting. Kocyigit et al. (2025) investigate the impact of data contamination on machine translation.

**Forgetting.** In machine learning, the term *forgetting* is frequently associated with *"catastrophic"* forgetting, where learning new tasks hurt the performance at previously solved tasks (Lopez-Paz & Ranzato, 2017). In the context of LLMs, catastrophic forgetting can occur during fine-tuning (Luo et al., 2023) or continual learning (Huang et al., 2024a). In contrast, this paper studies forgetting as a potential *"natural"* phenomenon of learning (Toneva et al., 2019). Tirumala et al. (2022) study forgetting in language modeling and find, similar to Toneva et al. (2019), that forgetting can be exponentially slow. In contrast, Jagielski et al. (2023) find that models empirically do forget examples over time. Pagliardini et al. (2025) propose to add a second momentum term to the AdamW optimizer, and show that this slows down the forgetting of past gradients.

**Data Attribution.** Data attribution methods (Koh & Liang, 2017; Ilyas et al., 2022; Park et al., 2023) aim to identify data points responsible for specific model behaviors. We ask how much a model's benchmark performance is influenced by seeing the example during training, which broadly falls within this field (Grosse et al., 2023; Choe et al., 2024). Importantly, we directly measure the influence of contaminated examples through explicit retraining, avoiding the approximation errors that can occur when using data attribution methods (Koh & Liang, 2017; Ghorbani & Zou, 2019) for large-scale models (Basu et al., 2021; Bae et al., 2022)

## 3. Background and Methods

This Section gives additional details on the research questions and lays out our experimental setup.

> **Research question**
>
> How does the presence of a text in the training data influence the final model's performance *on that same text*?

While it is well known that an individual data point can be influential if the training data and model are small (Koh & Liang, 2017), we are concerned with the large-data regime where the influence of any individual example may vanish (Basu et al., 2021; Bae et al., 2022). To further clarify this setup, Section 3.1 gives a brief overview of recent developments in scaling the training data of LLMs. Section 3.2 details the used benchmarks and explains how we contaminate the training data. Section 3.3 discusses the problem of near-duplicate benchmark questions.

**Models and Training Data.** We train language models of up to 1.6B parameters using the architecture and hyperparameters from the GPT-3 paper (Brown et al., 2020, Table 2.1). For this, we adopt the llm.c codebase. The training data is the 100BT split of the FineWeb-Edu dataset (Lozhkov et al., 2024). We also train OLMo-7B (Groeneveld et al., 2024) using the corresponding code and data (Soldaini et al., 2024).

### 3.1. We Consider the Regime of n-times Chinchilla

According to the Chinchilla scaling law, for every doubling of model size, the number of training tokens should also be doubled. The Chinchilla model itself has 70 billion (B) parameters and was trained on 1.4 trillion (T) tokens; suggesting that the number of training tokens should be roughly 20x the number of model parameters (Hoffmann et al., 2022, Table 3). While the Chinchilla paper was highly influential, modern language models are trained on significantly more tokens (Sardana & Frankle, 2024). For example, the OLMo-7B model was trained on 2.46T tokens, 17.5x the amount suggested by Chinchilla (Groeneveld et al., 2024). Similarly, the Llama 3 70B model was reportedly trained on 15T tokens, at over 10x Chinchilla (Dubey et al., 2024; Meta AI, 2024). The same holds for almost all recent 7B-parameter LLMs (Gemma Team, 2024). In this paper, *we count the number of tokens a model is trained on as a multiple of its Chinchilla tokens*.

### 3.2. We Evaluate on a Mix of Different Benchmarks

We evaluate the impact of data contamination using a *mix* of seven different benchmarks: ARC-Easy (Clark et al., 2018), Social-I-QA (Sap et al., 2019), WinoGrande (Sakaguchi

et al., 2021), PiQA (Bisk et al., 2020), BoolQ (Clark et al., 2019), MMLU (Hendrycks et al., 2021), and HellaSwag (Zellers et al., 2019). This means that every evaluation contains questions from all seven benchmarks. To construct the mixed contamination data, we first concatenate the different benchmarks. We then partition the set of all benchmark questions into subsets ranging from 10,000 to 2,000 questions so that each subset contains all benchmarks in equal weight: HellaSwag: 19.58%, SocialIQA: 8.27%, PiQA: 19.7%, MMLU: 21.82%, BoolQ: 6.48%, ARC-Easy: 5.92%, and WinoGrande: 18.16%. A holdout set of 10,000 benchmark questions is never added to the training data. The other subsets are added to the training data, repeated either 4, 12, 36, or 144 times.[2] We consider *exact* contamination, that is we contaminate the training data with the same texts that the model is later evaluated on. We insert benchmark questions *individually* and at *random* positions into the training data. Models are evaluated zero-shot via the likelihood assigned to different sentence completions (Gao, 2021). For more discussion and details about how contamination is performed, see Supplement A.3 and Supplement A.4.

### 3.3. We Filter Near-Duplicate Benchmark Questions

Our method requires no side effects from contaminating the training data with one question on the evaluation of another question. However, upon closer inspection, it turns out that *all* the commonly used benchmarks from the literature contain questions that are either near-duplicates or where the context of one question contains the answer to another question (for example, because the same text document was used to create multiple questions). To address this problem, we perform extensive filtering, removing duplicate questions where the length-normalized Levenshtein distance falls below a certain threshold (Levenshtein, 1966; Navarro, 2001). The issue of duplicate benchmark questions is detailed in Supplement A.5, where we also describe an experiment to verify that near-duplicate questions do not invalidate our method.

## 4. Experimental Results

We now present our main experimental results. We begin in Section 4.1 by discussing the scaling in model parameters, training tokens, and repetitions in the training data. The following Section 4.2 discusses various experiments on forgetting. The first two sections rely on training small GPT-3

---

[2]In preliminary experiments, we found that these numbers pleasantly cover the range from statistically significant contamination to complete overfitting. We also considered repeating observations a single time, as in (Jiang et al., 2024). However, we found this often leads to accuracy differences of about one or two percentage points, just within the margin of our confidence intervals, which is undesirable.

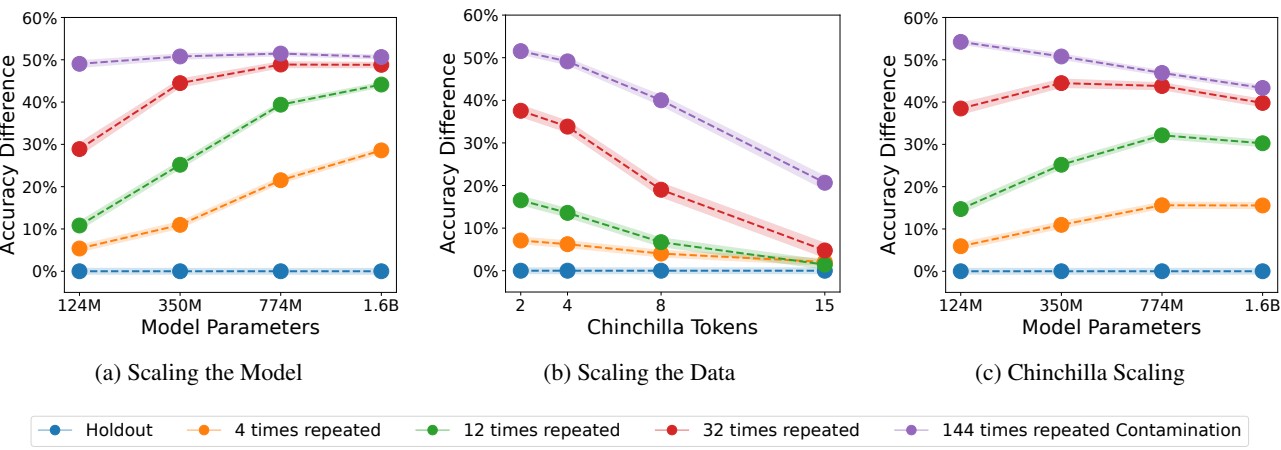

(a) Scaling the Model  (b) Scaling the Data  (c) Chinchilla Scaling

*Figure 1.* **Benchmark overfitting due to contamination.** **(a)** We train different models on 7B tokens. **(b)** We train 124M parameter models on increasingly many tokens. **(c)** We train models according to the Chinchilla scaling laws. The figure depicts the accuracy difference in percentage points between the holdout (normalized to zero) and the contaminated examples. The results are across a mix of seven different benchmarks, as outlined in Section 3.2. Different colors indicate different levels of contamination. Mean and bootstrapped 90% confidence intervals.

*Table 1.* Accuracies of the Chinchilla-optimal models. The table depicts the absolute accuracies of the Chinchilla-optimal models for holdout and contaminated benchmark questions.

| Model | Holdout | 4x | 12x | 32x | 144x |
|-------|---------|-------|-------|-------|-------|
| 124M | 42.22 | 48.14 | 56.92 | 80.70 | 96.45 |
| 350M | 44.72 | 55.69 | 69.90 | 89.20 | 95.50 |
| 774M | 49.16 | 64.76 | 81.30 | 92.95 | 96.05 |
| 1.6B | 52.06 | 67.61 | 82.32 | 91.85 | 95.40 |

models. Section 4.3 complements this with an analysis of OLMo-7B (Groeneveld et al., 2024).

### 4.1. Contamination Scales with Model, Data, and Repetitions

We conduct three different experiments to understand how the effect of data contamination scales with the number of model parameters, training tokens, and the number of times a contaminated example is seen. First, we train increasingly large models on the same dataset of 7B tokens. Second, we train 124M parameter models on increasingly many tokens. Third, we train increasingly large models according to the Chinchilla scaling laws (Hoffmann et al., 2022), meaning that the number of training tokens scales linearly with the model parameters. In all experiments, we contaminate the training data *uniformly at random* with benchmark questions.

Figure 1 depicts the results of all three experiments. Because we are interested in the performance *difference* between the holdout data and the contaminated examples, Figure 1

depicts the *accuracy difference* between the holdout and contaminated examples in percentage points. In Figure 1a, we see that the accuracy difference due to contamination is *increasing in the number of model parameters*. For a 124M parameter model trained on 7B tokens, the overfitting due to 4 times contamination is 5 percentage points. For a 1.6B parameter model train on the same dataset, it is 20. Next, Figure 1b shows that the accuracy difference is *decreasing in the number of training tokens*. For a 124M parameter model trained at 2x Chinchilla, the accuracy difference due to 12 times contamination is 18 percentage points. For a 124M parameter model trained at 15x Chinchilla, the respective accuracy difference is within the confidence interval of the holdout. From Figure 1, we also see that the accuracy difference is *increasing in the number of times an example is repeated*. For a 350M parameter model trained on 7B tokens, the accuracy difference is 11, 25, 44, and 51 percentage points for 4, 12, 32, and 144 times repeated contamination, respectively.

Because the accuracy difference *increases* in the number of model parameters and *decreases* in the number of tokens, the interesting question is how it behaves if model parameters and tokens are scaled *jointly*. A natural starting point is to double the number of training tokens for every doubling of model parameters, as specified by the Chinchilla scaling laws (Hoffmann et al., 2022). Figure 1c depicts the accuracy difference due to contamination as we train increasingly large Chinchilla-optimal models. While there is no clear monotone pattern, we see that moderate amounts of contamination can lead to significant overfitting. For the 774M parameter model, 4 times repeated contamination leads to an accuracy difference of 15 percentage

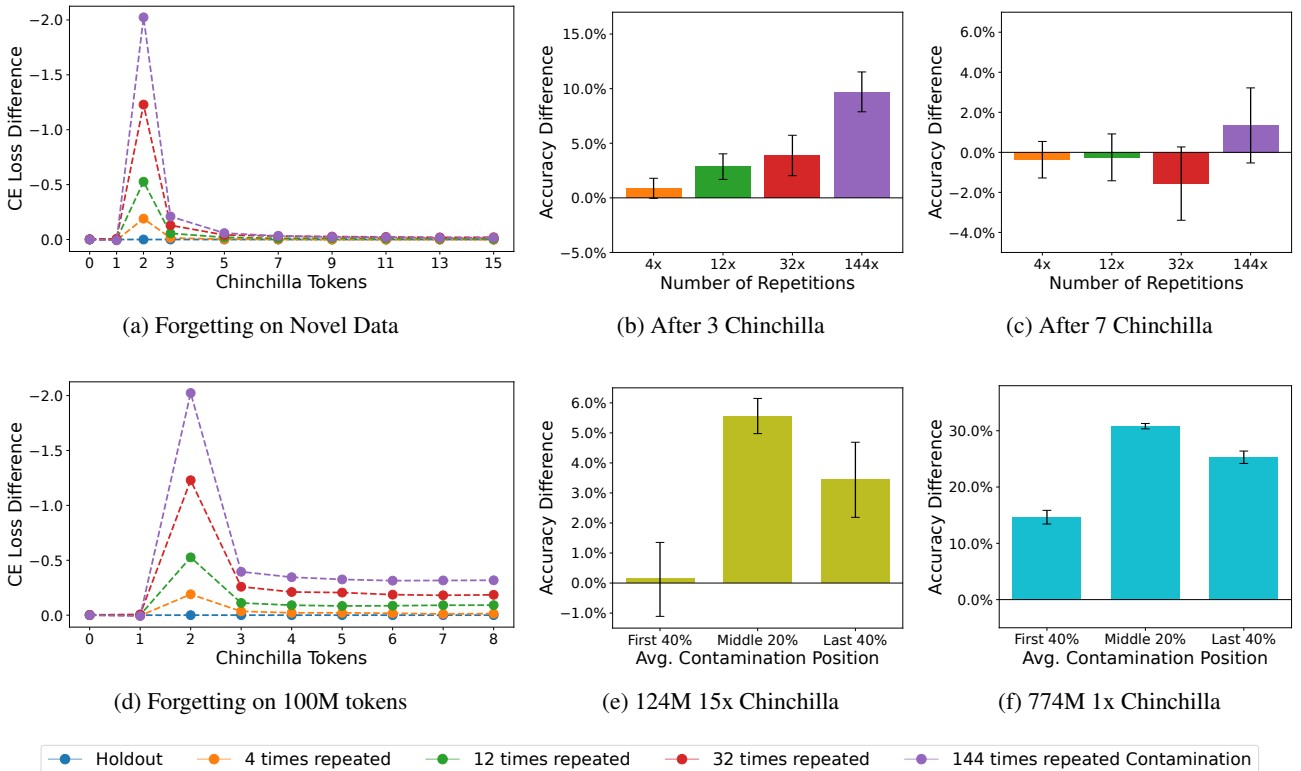

*Figure 2.* **The forgetting dynamic of neural network training.** **(a)** The development of the cross-entropy loss difference between contaminated and holdout benchmark questions over the course of training. Contamination occurs between the first and second Chinchilla (1 and 2 on the x-axis). **(b)** Accuracy differences after training for 3 Chinchilla. **(c)** Accuracy differences after training for 7 Chinchilla. **(d)** Same as (a). **(e)+(f)** The accuracy difference depends on the average position of an example in the training data. Mean and bootstrapped 90% confidence intervals.

points, suggesting that *under Chinchilla training, a single time of contamination can lead to overfitting of as much as 3 percentage points.*[3]

## 4.2. Contamination Can be Completely Forgotten

In the previous Section 4.1, we saw that the accuracy difference due to contamination decreases in the number of tokens up to the point where even 12 repetitions of a benchmark question in the training data can become insignificant. In this Section, we identify the forgetting dynamic of neural network training as the reason for this effect. We discuss how quickly forgetting occurs, whether examples are completely forgotten, and what kind of repetition makes a model remember.

To study the effect of forgetting, we train a 124M parameter model at 15x Chinchilla. Instead of contaminating

uniformly over the course of training like in the previous Section 4.1, we perform the contamination between the first and second Chinchilla.[4] Figure 2a depicts the development of the *difference* in cross-entropy loss between contaminated and clean benchmark questions over the course of training. We see a strong peak after 2 Chinchilla, which is expected and shows the effect of contamination. What is interesting to us is the rate at which the cross-entropy loss difference decays as we continue training. After training for 1 additional Chinchilla (2.5B tokens for the 124M parameter model), it has already decayed significantly. However, the difference is still visible in Figure 2a. Figure 2b depicts the corresponding accuracy differences at this point, and we see that all contamination levels still lead to overfitting. As we continue training, the cross-entropy loss difference between contaminated and holdout questions further narrows.

---

[3]To contaminate the training data of a 774M model a single time with 10,000 benchmark questions, we need to insert ~0.5 million tokens into 15.5 billion training tokens, about 0.003% of the training data.

[4]Note that the model is already fairly trained after the first Chinchilla, meaning that the contamination is not very early during training. This is important because there is evidence that observations are more quickly forgotten if the model has not yet learned representations (Jagielski et al., 2023; Cao et al., 2024; Huang et al., 2024b). This is *not* the setting we are studying here.

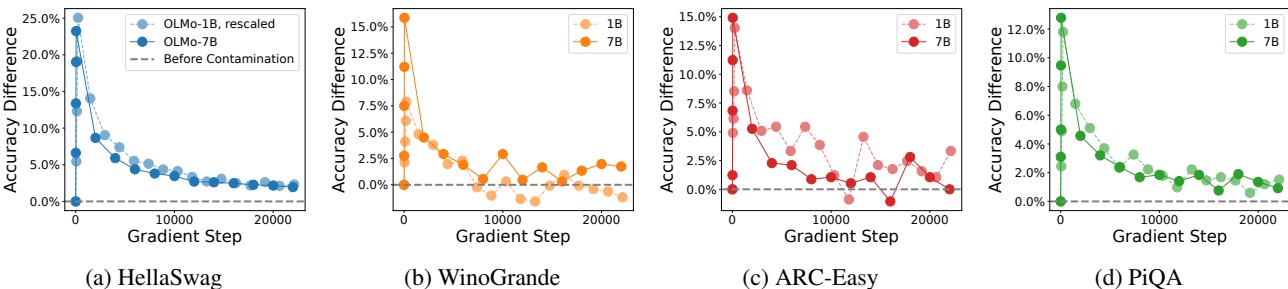

(a) HellaSwag      (b) WinoGrande      (c) ARC-Easy      (d) PiQA

*Figure 3.* **Contamination and Forgetting in OLMo-1B and OLMo-7B.** We contaminate intermediate OLMo checkpoints four times with different benchmarks. This causes an average accuracy increase of 17 percentage points for the 7B model. We then continue pre-training for 13% of the remaining training time (or 25000 gradient steps), leading to a reduction of 87% of the accuracy increase due to contamination. The solid line depicts the result for OLMo-7B, and the dashed line depicts the results for OLMo-1B. To account for the difference in the number of model parameters, we scaled the curve of the 1B parameter model by a factor of 5.9 (the actual parameter ratio between the two models), after which the forgetting curves of both models align remarkably well (the plot depicts approximately 3500 gradient steps for the 1B model, and 25000 gradient steps for the 7B model). Different colors correspond to different benchmarks, and the grey line depicts the clean accuracy before contamination. Supplement Figures 9 and 10 depict the results for both models separately.

From Figure 2c, which depicts the accuracy differences after forgetting for a total of 5 Chinchilla, we see that the effect of contamination is eventually *completely forgotten* in the sense that there is no longer any accuracy difference between contamination and holdout benchmark questions.

The result that contamination can be completely forgotten is in contrast to some previous work on forgetting which have found that forgetting approaches a stable baseline Tirumala et al. (2022, Figure 10), or that certain examples are never forgotten (Toneva et al., 2019). To understand this difference, observe that many previous works on forgetting have not trained on a continuous stream of data. Instead, they have trained on the same training set for multiple epochs. Consequently, we modify our forgetting experiment to repeatedly train on the same 100M tokens after the second epoch. The result of this experiment is depicted in Figure 2d and should be compared to Figure 2a. Interestingly, this simple modification causes the effect of forgetting to stabilize at a level strictly larger than zero. We conclude that *exposure to novel data is important for forgetting*, an observation similar to Jagielski et al. (2023).

To further understand the impact of forgetting, we now ask whether examples seen late during training influence model behavior more strongly than examples seen early during training. To study this question, we average all the different *uniform* contamination levels from the models in the previous Section 4.1 (to gain statistical power) and consider the amount of overfitting depending on whether a question is seen, on average, in the beginning, middle, or end of training. The result of this experiment is depicted in Figure 2e and Figure 2f. As expected under forgetting, we see that benchmark questions seen early during training exhibit the smallest amount of overfitting. Interestingly and somewhat unexpectedly, questions that are neither clustered

towards the beginning nor the end but as uniformly distributed throughout training as possible exhibit the strongest overfitting, suggesting that this spaced form of repetition helps the model remember (the middle peak is the most pronounced both in Figure 2e and Figure 2f).

### 4.3. Contamination and Forgetting in OLMo

In the previous sections, we trained small GPT-3 models from scratch. In this Section, we complement this analysis by continual pre-training from intermediate OLMo-1B and OLMo-7B checkpoints (Groeneveld et al., 2024). Similar to the analysis in Section 4.2, we insert the benchmark data at a specific point into the training data and then measure the subsequent forgetting. Unlike in the previous Section, we now insert the entire benchmark data – we already have a "clean" baseline from the original OLMo training runs. We insert each benchmark question four times, and contaminate with four different benchmarks: HellaSwag (Zellers et al., 2019), WinoGrande Sakaguchi et al., 2021, ARC-Easy (Clark et al., 2018), and PiQA (Bisk et al., 2020).

Figure 3 depicts the result of the experiment. The leftmost point in every plot corresponds to the uncontaminated model, and the following four points depict the effect of contamination. The plots then depict how the effect of contamination is forgotten as we continue training. Similar to the results in the previous sections, *the immediate effect of contamination is significant*, leading to an average accuracy increase of 17 percentage points across the different benchmarks for the 7B model. At the same time, *the effect of contamination decays considerably as we continue training*. After continual pre-training for 13% of the remaining training time, the evaluation on WinoGrande and ARC-Easy for the 7B model is no longer significantly different from the uncontaminated evaluation. On HellaSwag and PiQA, the effect of contam-

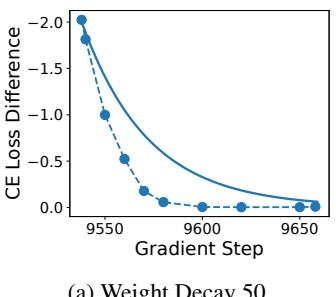 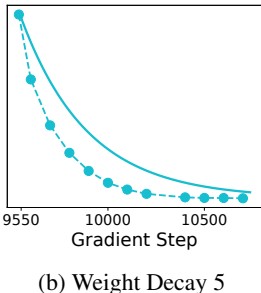 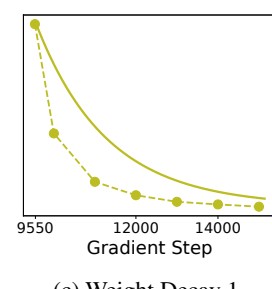 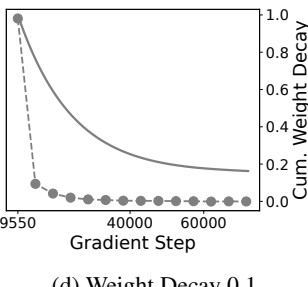

(a) Weight Decay 50     (b) Weight Decay 5     (c) Weight Decay 1     (d) Weight Decay 0.1

*Figure 4.* **Empirical forgetting occurs faster than the cumulative weight decay.** We continue training the contaminated 124M parameter model from Section 4.2. The figure depicts empirical forgetting (dashed line) and the cumulative weight decay (solid line) for four different choices of the weight decay parameter. Forgetting depends strongly on the weight decay parameter, as is evident from the very different **x-axis scales**, ranging from 120 gradient steps in (a) to 62500 gradient steps in (d).

ination has decayed to approximately 2 percentage points of overfitting. This demonstrates that *the effect of forgetting during pre-training is significant, even for a 7B parameter model.*

Interestingly, we find that forgetting in OLMo exhibits a scaling behavior. To see this, note that Figure 3 also depicts the result of the forgetting experiment with OLMo-1B. However, we rescaled the forgetting curve of the 1B parameter model by a factor of 5.9, meaning that at any given point on the x-axis, the 7B parameter model has seen 5.9 times as many tokens as the 1B parameter model.[5] We choose the factor 5.9 because it corresponds to the actual parameter ratio between the two models (OLMo-1B has 1,176,764,416 parameters, whereas OLMo-7B has 6,888,095,744 parameters). With this rescaling, the forgetting curves of the two models are surprisingly well aligned, suggesting that forgetting with a model that has 5.9 times as many parameters requires 5.9 times as many tokens. Of course, this observation is well aligned with the results on scaling in Section 4.1.

## 5. Weight Decay and Forgetting

In the previous Section 4, we have seen that forgetting is an important empirical property of LLM training. Which factors contribute to the phenomenon? In this section, we show that the weight decay parameter and learning rate schedule of the AdamW optimizer play a part in forgetting past training examples. This offers a novel perspective on the weight decay parameter, usually seen in terms of generalization and training stability (Van Laarhoven, 2017; Zhang et al., 2019; Lewkowycz & Gur-Ari, 2020; Andriushchenko et al., 2024).

---

[5]Here we refer to the number of tokens seen in the experiment. The total number of tokens seen by the two models during pre-training up to this point is approximately the same.

### 5.1. The Cumulative Weight Decay

Consider the parameter update of AdamW at gradient step $t \geq 1$. It consists of two decoupled updates: A weight decay update given by

$$\hat{\theta}_t = \theta_{t-1} - \gamma_t \lambda \theta_{t-1}, \tag{1}$$

and a gradient update given by

$$\theta_t = \hat{\theta}_t - \gamma_t \hat{m}_t / (\sqrt{\hat{v}_t} + \epsilon). \tag{2}$$

Here, $\theta_t$ are the model parameters, $\gamma_t$ is the learning rate, $\lambda$ is the weight decay parameter, and $\hat{m}_t$ and $\hat{v}_t$ are first- and second-order moment estimates of the gradient (Loshchilov & Hutter, 2019; PyTorch Contributors, 2024). Denoting the model weights at initialization by $\theta_0$, and the adaptive gradient by $\hat{g}_t = \hat{m}_t / (\sqrt{\hat{v}_t} + \epsilon)$, we can iterate (1) and (2) to obtain

$$\theta_T = w_0^T \theta_0 - \sum_{t=1}^{T} w_t^T \gamma_t \hat{g}_t \tag{3}$$

where the *cumulative weight decay* is computed as follows

$$w_{t_1}^{t_2} = \prod_{i=t_1+1}^{t_2} (1 - \gamma_i \lambda). \tag{4}$$

Equation (3) shows that the model weights after $t$ gradient steps are a weighted average of the initial model weights and all the adaptive gradient updates up to time step $t$. In other words, the weights $w_{t_1}^{t_2}$ determine the influence of the gradient update at time $t_1$ on the model weights at time $t_2$. As recently observed by Wang & Aitchison (2024), the weights $\theta_T$ can also be understood as an exponential moving average of past gradient updates. With weight decay, $w_{t_1}^{t_2}$ decays to zero, as the following proposition describes.

*Proposition* 1. (Cumulative Weight Decay) Fix $\epsilon > 0$. Let $T = t_2 - t_1$ be the number of gradient steps that have passed since step $t_1$. Let $\gamma_{\text{avg}} = \frac{1}{T} \sum_{t_1+1}^{t_2} \gamma_i$ be the average learning rate between gradient steps $t_1$ and $t_2$. If $T$ is sufficiently

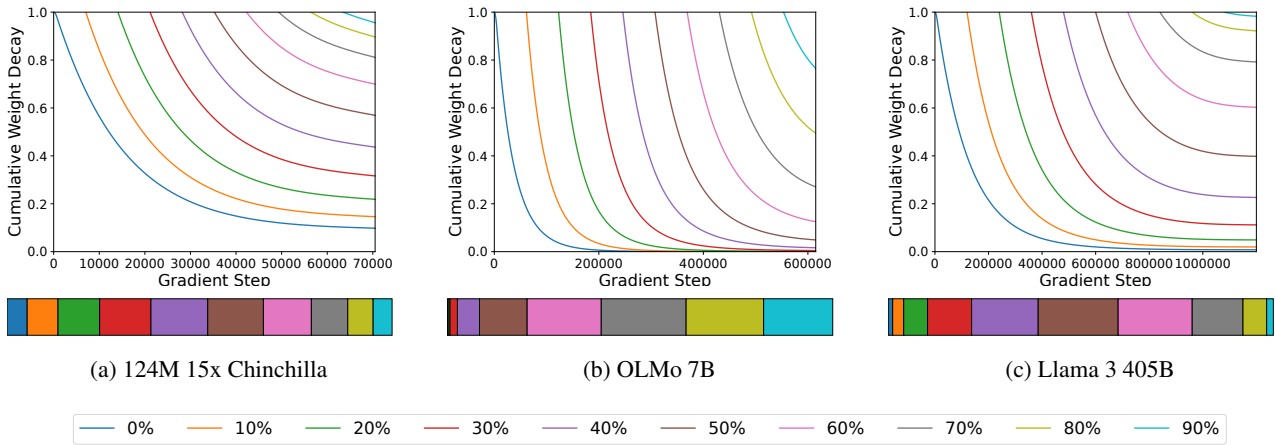

(a) 124M 15x Chinchilla  (b) OLMo 7B  (c) Llama 3 405B

| — 0% | — 10% | — 20% | — 30% | — 40% | — 50% | — 60% | — 70% | — 80% | — 90% |

*Figure 5.* **Cumulative weight decay and approximate model weight composition for different large-scale training runs.** *Top Row:* The cumulative weight decay $w_{t_1}^{t_2}$ as defined in equation (4) for different training runs. The figures depict the decay of the gradient updates for every decile of the training run, indicated in different colors. *Bottom Row:* The approximate composition of the final model weights in terms of the gradient updates from different deciles of the training run. The deciles are indicated in colors depicted in the legend below the plot. The cumulative weight decay can be easily computed using this Jupyter Notebook.

large, that is $T \geq \frac{\log(1/\epsilon)}{\lambda \gamma_{\text{avg}}}$, then

$$w_{t_1}^{t_2} \leq \epsilon.$$

We present a proof in Supplement A.8.

What is the relationship between cumulative weight decay and forgetting? The adaptive gradient $\hat{g}_{t_1}$ contains the relevant information about the training examples seen at time $t_1$. If $\hat{g}_{t_1}$ is no longer part of the model weights because $w_{t_1}^{t_2}$ has decayed to zero, then the model should no longer depend on the examples seen at time $t_1$. In other words, the training examples seen at time $t_1$ should be forgotten. However, the data seen at time $t_1$ still influences $\theta_T$ via the trajectory of gradient descent, even if the direct contribution of $\hat{g}_{t_1}$ has decayed to zero. Because of this, we now investigate the relationship between forgetting and weight decay experimentally.

### 5.2. Empirical Forgetting Occurs Faster than the Cumulative Weight Decay

We now study the causal effect of weight decay on forgetting by intervening on the weight decay parameter, keeping everything else constant. We consider the contaminated model from Section 4.2 after two Chinchilla and continue training with four different choices of the weight decay parameter. Figure 4 depicts the result of this experiment. Every plot in Figure 4 depicts two curves. The cumulative weight decay is depicted as a solid line. The empirically observed forgetting for 144 times repeated contamination is depicted as a dashed line (compare Figure 2a). From Figure 4, we see that increasing the weight decay parameter increases

the empirical rate of forgetting. This is evident from the very different x-axis scales in the different plots. Moreover, the dashed line always lies below the solid line, meaning that *empirical forgetting occurs faster than the cumulative weight decay*. This is despite the fact that weight decay is not necessary for forgetting (see Supplement Figure 11).

### 5.3. Cumulative Weight Decay in Large-Scale Training

Given Proposition 1 and the experimental results in Section 5.2, we hypothesize that example forgetting occurs at least as fast as the cumulative weight decay. Based on this hypothesis, we now gauge the degree of forgetting in large-scale training runs. The key insight here is that the cumulative weight decay depends only on the learning rate schedule and the weight decay parameter. This means that we can study the cumulative weight decay even in large-scale training runs that far exceed our experimental budget. Ideally, we would of course want experimental evidence on forgetting in large-scale training, akin to the analysis in Section 4.1. Since this is not possible within our experimental budget, we resort to a heuristic analysis of example forgetting via the cumulative weight decay.

Figure 5 depicts the cumulative weight decay $w_{t_1}^{t_2}$ for three different models: The 124M parameter model from Section 4.2, OLMo 7B (Groeneveld et al., 2024), and Llama 3 405B (Dubey et al., 2024) (where we assume the model trained with a weight decay of 0.1). The figure depicts the evolution of the term $w_{t_1}^{t_2}$ for ten fixed choices of $t_1$, one at each decile of the training run. As discussed above, we can interpret the curves in Figure 5 as heuristic *forgetting curves*

that indicate the degree to which the data seen at different stages is forgotten as we continue training. For the 124M model depicted in Figure 5a, even the initialization is not completely forgotten at the end of training (the blue curve is still significantly larger than zero). For OLMo-7B, however, the weights of the gradients of the first 40% of training decay to zero until the end of training, meaning that the data seen during this time is likely forgotten (Figure 5b). The same holds for Llama 3 405B, where the first 10% of the training data is likely forgotten (Figure 5c).

By integrating $w_{t_1}^T$, the cumulative weight decay even allows us to approximate the composition of the final model weights in terms of the gradient updates from different deciles of training. This is depicted in the bottom row of Figure 5.

## 6. Discussion

In this work, we have connected the literature on data contamination (Jiang et al., 2024) with research on forgetting (Tirumala et al., 2022; Jagielski et al., 2023). Through careful controlled experiments, we have shown that the impact of contamination depends on the joint scaling of model, data and contamination — an aspect that has largely been overlooked in the existing literature (Yang et al., 2023; Oren et al., 2024; Jiang et al., 2024). Perhaps surprisingly, we have also seen that the impact of contamination can be completely forgotten, an effect that is especially relevant for training runs that operate in the large-data regime (Groeneveld et al., 2024; Gemma Team, 2024). As a consequence, large-data LLM pre-training also appears to be empirically *stable* with respect to individual benchmark questions (Bousquet & Elisseeff, 2002). Interestingly, we have also seen a number of ways in which LLM pre-training is different from other learning setups. For example, we have seen that forgetting behaves very differently when training on a continuous stream of novel data as opposed to multi-epoch training on a limited dataset (Tirumala et al., 2022; Muennighoff et al., 2023).

The *"contamination problem"* as studied in this paper has interesting connections to many areas of machine learning, including privacy (Graves et al., 2021; Jagielski et al., 2023), data attribution (Kirchenbauer et al., 2024), and generalization (Bousquet & Elisseeff, 2002; Hardt et al., 2016; Mania et al., 2019). The connection to data attribution arises from the fact that the presence or absence of a benchmark question in the pre-training data sometimes appears to be irrelevant for the model behavior *on that same datapoint*, suggesting that it does not make sense to attribute model behavior to individual datapoints in this regime.

An important limitation of our work is that our empirical results on forgetting are restricted to benchmark questions.

This means that one has to be careful when extrapolating our results to other contexts, especially to a privacy setup (Carlini et al., 2019; Jagielski et al., 2023). This is because empirical forgetting might behave differently for random strings or otherwise uniquely identifiable information (Carlini et al., 2021).

## Impact Statement

This paper studies the empirical properties of training deep neural networks. Because our study concerns the scientific question of model evaluation, we don't believe it raises significant ethical concerns. That being said, contamination and memorization have broader implications, including copyright and privacy.

## Acknowledgments

We would like to thank Jonas Geiping and Dirk Groeneveld for helpful discussion about data contamination. This work is supported by the Tübingen AI Center, the German Research Foundation through the Cluster of Excellence "Machine Learning - New Perspectives for Science" (EXC 2064/1 number 390727645), and the CZS Institute for Artificial Intelligence and Law.

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

# A. Appendix

## A.1. Takeaways

Here we list some important takeaways from our work.

1. The impact of data contamination is not fixed, but depends on the scale of model, data, and contamination.

2. If the scale of the data is large compared to the scale of the model and contamination, then the causal effect of contamination can be zero.

3. The interplay of two opposing mechanisms determines the impact of seeing a benchmark question during pre-training. Directly after seeing the question, there is a spike in the likelihood that corresponds to overfitting. Subsequently, there is a forgetting effect, meaning that the impact of the text on the model decays as we continue training.

4. LLM pre-training is fairly stable with respect to individual benchmark questions.

5. LLM pre-training, where the model is continuously exposed to novel data over many gradient steps, differs from multi-epoch training and small-data fine-tuning setups because of forgetting.

6. The weight decay parameter in the AdamW optimizer has a causal effect on forgetting.

7. Cumulative weight decay might be a useful heuristic for forgetting in large-scale training runs.

## A.2. Limitations

Here we list some important limitations of our work.

1. The experiments in this paper were conducted only with benchmark questions, and only for the setup of exact contamination.

2. Forgetting might behave differently for random strings or other uniquely identifiable information.

3. Further experiments would be required to validate our results at a larger scale ($> 7$B parameters).

## A.3. Additional Discussion of Data Contamination Assumptions and Setting

Here, we discuss our data contamination approach in a bit more detail.

In this paper, we consider only **exact contamination**. This means we contaminate the training data exactly with the text the model is later evaluated on. In the literature, it has been shown that non-exact contamination (re-worded questions, translation into a different language) can affect benchmark performance, too. For example, Yang et al. (2023) have shown that a 13B parameter Llama 2 Model (Touvron et al., 2023) can achieve an accuracy increase of over 20 percentage points after training on re-phrased benchmark questions. We decided against considering non-exact contamination in this paper because the models we train from scratch are much smaller than those for which non-exact contamination results have been shown. This means these models are less capable of making sense of related information, potentially leading us to underestimate the effect of non-exact contamination for realistic training runs.

In addition, we consider contamination with **individual benchmark questions**, inserted into the training data at **random** positions. We consider this setup because we are interested in contamination from the perspective of *leakage*, where individual benchmark questions may enter the training data via different documents (for example, as quotes in Wikipedia articles, a case described in Brown et al. (2020)). This contrasts with the setup where a dataset is present in the training data as a long contiguous string, which we conjecture might have a similar impact but be easier detectable (Oren et al., 2024). The fact that we contaminate with benchmark questions also sets us apart from related works that study data contamination and memorization for random strings and uniquely identified objects (Carlini et al., 2019; 2021). It is worth highlighting that the results between these two setups might differ, especially considering the time it takes to forget an example.

We only consider pre-training.

*Table 2.* Overview of benchmarks used in the paper. This table documents the experiments with GPT-3 models. The first two rows provide the dataset split and corresponding number of benchmark questions. The third row provides the number of questions that were removed from the dataset after filtering each dataset for near-duplicate questions. The fourth row provides the number of questions that were removed after additionally filtering for near-duplicate questions across all the different datasets combined. The fifth row provides the dataset's weight in the dataset splits used in the experiments.

| | HellaSwag | PiQA | Social-i-QA | BoolQ | MMLU | WinoGrande | ARC-Easy |
|---|---|---|---|---|---|---|---|
| **Split** | Validation | Train | Train | Validation | Test | XL, Train | All |
| **Size** | 10,042 | 16,113 | 33,410 | 3,269 | 14,042 | 40,398 | 5,197 |
| **Filtered** | 1,416 | 7,386 | 29,756 | 409 | 4,423 | 21,944 | 2,568 |
| **Cross-Filtered** | 3 | 6 | 10 | 2 | 15 | 0 | 13 |
| **Weight** | 19.58% | 19.77% | 8.27% | 6.48% | 21.82% | 18.16% | 5.92% |

---

**HellaSwag:** A woman stands holding a violin against herself. The woman  plays the violin. The woman stops playing the violin.

**HellaSwag:** A man is standing outside holding a violin. He begins to play the violin. he stops and sets the violin to his side.

**ARC-Easy:** Question: A student is playing with a small toy boat […] The boat moves toward the shore because the waves transfer Answer: energy.

**MMLU:** Question: A wave transfers Answer: energy

---

*Figure 6.* Language modeling benchmarks frequently contain near-duplicate questions. We perform extensive filtering for duplicates using fuzzy string matching. The figure depicts a near-duplicate from HellaSwag and a cross-benchmark duplicate from ARC-Easy/MMLU.

## A.4. Additional Details on Evaluation and How Contamination was Performed

**Benchmark Questions and Evaluation.** We use code from OLMo (Groeneveld et al., 2024) to format the different benchmark questions. This code is again based in part on the EleutherAI Evaluation Harness (Gao et al., 2024). The benchmark questions are multiple-choice, and the different options are presented to the model zero-shot as possible sentence continuations. The prediction is the sentence continuation with the largest likelihood. For the small GPT-3 models, we normalize by the number of tokens (Gao, 2021). For OLMo, we rely on the evaluation framework that is part of the code repository.

**Inserting benchmark questions into the training data.** A batch of LLM training data consists of $B$ sequences of $S$ tokens, resulting in a batch size of $B \times S$. For example, OLMo-1B is trained with $B = 2048$ and $S = 2048$; the batch for a single gradient step contains ~4M tokens (Groeneveld et al., 2024). Individual sequences in a batch usually contain multiple texts separated by a special end-of-text token. We insert benchmark questions at random positions into the pre-training data, separated at the beginning and end with the end-of-text token.

## A.5. Filtering Near-Duplicate Benchmark Questions

Upon close inspection of the different benchmarks, it turns out that there are exact and near-duplicate questions both within and across benchmarks. Consider, for example, the following two examples from HellaSwag (Zellers et al., 2019):

**HellaSwag 1:** *A person is seen riding a board along the water with a kite on top. more clips are shown of the person riding back and fourth on the board.*

and

**HellaSwag 2:** *A person is seen riding a board along the water with a kite on top. More clips are shown of the*

*person riding back and fourth on the board. the person continues to ride the board along the water.*

Note that these are the ground-truth options of two *different* benchmark questions. Similar patterns can be observed for many benchmarks, for example, because a single text document was used to create different benchmark questions. Here is another example from PiQA (Bisk et al., 2020):

> **PiQA 1:** *Goal: how do you close a cabinet? Solution: push the door shut.*

and

> **PiQA 2:** *Goal: how do you close a cabinet? Solution: shut the door.*

Near-duplicate benchmark questions present a challenge for our methodology. This is because one question might end up in a set of benchmark questions that we highly contaminate with, whereas the other question might end up in the purportedly "clean" set of benchmark questions that we use for holdout evaluation. To tackle this problem, we perform fuzzy string matching between the ground-truth options (that is, the potential contamination data) of all benchmark questions, randomly removing one question for every detected duplicate. We use the Python package rapidfuzz.

**Summary Statistics.** Table 2 depicts summary statistics about the different benchmarks, including the number of questions that were filtered during the duplicate-detection stage. We see that the number of filtered questions is significant. On some datasets, especially Social-i-QA, we had to apply very aggressive filtering to avoid any side-effects during contamination. Hence, the number of removed questions per dataset does not necessarily reflect the actual number of duplicates, but the level of filtering that had to be applied to remove all duplicate questions.

**Experimental verification that filtering worked.** We verify that our filtering procedure worked by training two models: One that is heavily contaminated (obtaining an accuracy of over 97%), and another model that did not see any contamination. We then evaluate both models on a set of 10,000 benchmark questions that are holdout even for the contaminated model. The contaminated model obtains an accuracy of 42.2%, (95%-CI: 41.2% - 43.2%) on the holdout, while the clean model obtains an accuracy of 41.9% (95%-CI: 41.0% -42.9%). Because the observed accuracy difference is small in absolute terms and lies within the confidence interval, we conclude that there are no significant side-effects in our evaluation procedure.

## A.6. Measuring the Overlap Between the Benchmark Questions and the Pre-Training Data

Here, we analyze the overlap between the 44000 de-duplicated benchmark questions and the pre-training data. Note that we did not de-contaminate the pre-training data (we use FineWeb-Edu "as-is"). In addition, note that our experimental design is valid even if the pre-training data is already contaminated (if there is already contamination, our experiments measure the causal effect of inserting the benchmark questions $k$-additional times). Our implementation of fuzzy string matching used to de-duplicate the benchmark questions is not efficient enough to match all benchmark questions against the pre-training data. Hence, we follow Singh et al. (2024) and report the length of the longest common substring between a benchmark question and the pre-training data. For simplicity, we report the overlap between all benchmark questions and the entire 100B token split of FineWeb-Edu. In our experiments, the training data is a random subset of these 100B tokens.

Figure 7 depicts the overlap between the benchmark questions and the pre-training data. For every benchmark question, we compute the length of the longest common substring divided by the length of the benchmark question (that is, a value of 1 means that the benchmark question occurs verbatim in the pre-training data). Figure 7 depicts the result of this computation in aggregated form. From Figure 7, we see that there is little overlap between the pre-training data and Hellaswag, PiQA, Social-i-QA, and Winogrande. For ARC-Easy, MMLU, and especially BoolQ, we find that there exist benchmark questions that have very high overlap with the pre-training data. Depending on the format of the questions and the nature of the overlap, some of these cases essentially correspond to verbatim contamination. In other cases, there might be a high overlap between the benchmark question and the pre-training data, but it is not necessarily clear to what degree the text in the pre-training data actually reveals the true answer to the benchmark question. On BoolQ, for example, many of the overlaps with the pre-training data occur for the descriptive contexts of the questions, which are the same for all answer options. Or consider the following example from ARC-Easy, where the pre-training data contains all four answer options.

> **Arc-Easy:** *Question: In Colonial America, people used ice to help keep foods fresh. They cut the ice from lakes and ponds during the winter and stored the ice in ice houses. They sometimes used hay as an insulator to prevent*

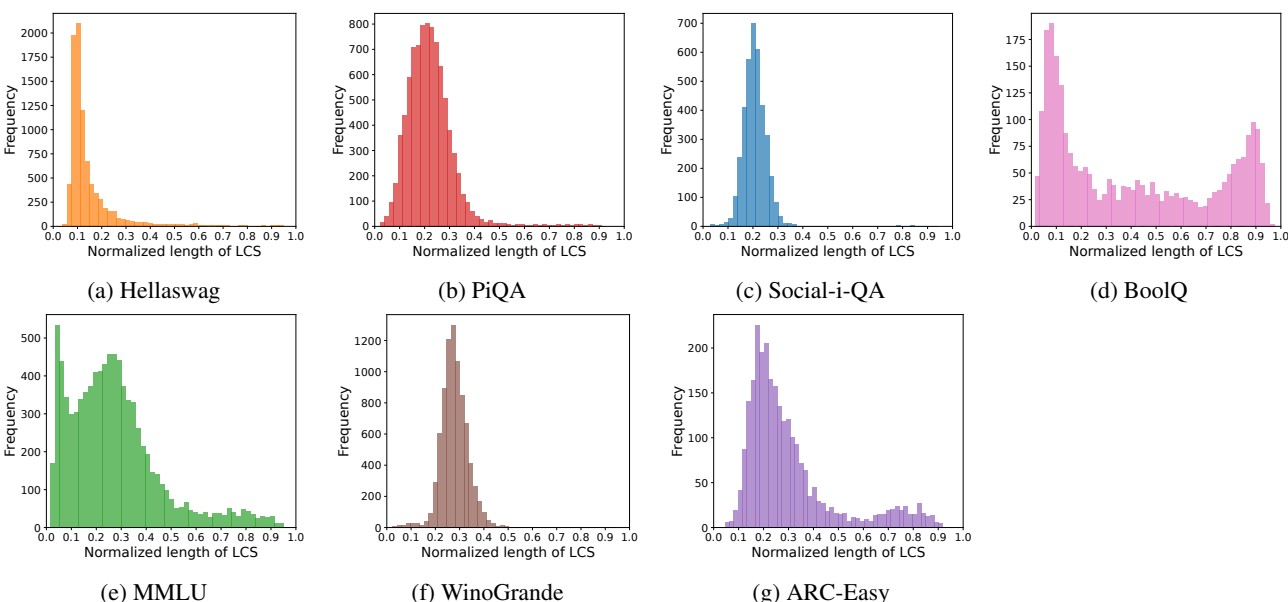

*Figure 7.* **Overlap between the benchmark questions and the 100B token split of FineWeb-Edu (Single Best Match).** For every benchmark question, we search in the pre-training data for the longest common substring. The figure depicts the distribution of the length of the longest common substring, normalized by the length of the benchmark question, for the different benchmarks.

> *the ice from melting. If you wanted to build an icehouse today, which of the following would be the best material to use as an insulator?*
> *Answer: foam blocks*

and

> **Pre-training data:** *In Colonial America, people used ice to help keep foods fresh. They cut the ice from lakes and ponds during the winter and stored the ice in ice houses. They sometimes used hay as an insulator to prevent the ice from melting. If you wanted to build an icehouse today, which of the following would be the best material to use as an insulator?*
> *(A) dried leaves*
> *(B) foam blocks*
> *(C) plastic wrap*
> *(D) rock salt*
> *In order to build a wooden table, [...]*

For the correct interpretation of Figure 7, note that is depicts the length of the *single best match* between the benchmark questions and the entire pre-training corpus of 100B tokens. For comparison, Figure 8 depicts the distribution of the length of the 10 best matches in the pre-training data. From Figure 8 (a), we see that searching for the 10 best matches does not affect the overall frequency distribution on Social-i-QA, for which there was no significant overlap in the first place. In contrast, for BoolQ, MMLU, and ARC-Easy, for which there was significant overlap, we observe that the frequency distribution of the 10 best matches shifts significantly to the left, indicating that regions of high overlap occur less than 10 times in the overall pre-training data.

In summary, we find that there are benchmark questions that have high overlap with the pre-training data, and that these questions are mostly from BoolQ, MMLU and ARC-Easy. At the same time, most of our 44000 benchmark questions do not have any significant overlap with the pre-training data as measured by the longest common substring metric. For those questions that have a high overlap, such an overlap usually occurs less than 10 times in the 100BT split of FineWeb-Edu.

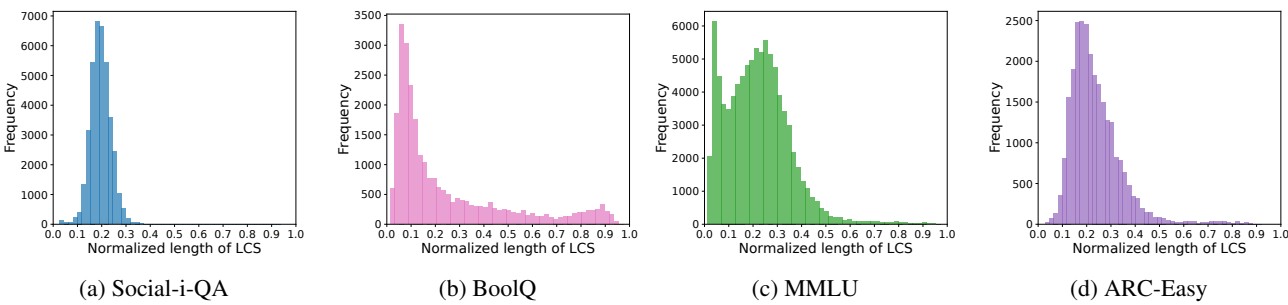

(a) Social-i-QA      (b) BoolQ      (c) MMLU      (d) ARC-Easy

*Figure 8.* **Overlap between the benchmark questions and the 100B token split of FineWeb-Edu (10 Best Matches).** For every benchmark question, we search in the pre-training data for the 10 longest common substrings. The figure depicts the distribution of the length of the longest common substrings, normalized by the length of the benchmark question, for the different benchmarks.

More research is needed to investigate different types of overlap between benchmark questions and the pre-training data, and whether they lead to benchmark overfitting (see also Jiang et al. (2024), who investigate n-gram-based overlap).

### A.7. Reproducibility

The code for this paper is available at github.com/tml-tuebingen/forgetting-contamination. It relies on the OLMo codebase, available at github.com/allenai/OLMo, and the llm.c codebase, available at github.com/karpathy/llm.c. Our code is fully reproducible, including the random positions in which the benchmark questions were inserted into the training data. We trained on the 100BT split of the FineWeb-Edu dataset, available at huggingface.co/datasets/HuggingFaceFW/fineweb-edu. Model training relied on Pytorch (Paszke et al., 2019) and was performed on 8xA100 nodes for all experiments except the continual pre-training of OLMo-7B, which ran for 6 weeks on 4xH100.

### A.8. Proof of Proposition 1

Fix $\epsilon > 0$. Let $T = t_2 - t_1$ be the number of gradient steps that have passed since step $t_1$. Let $\gamma_{\mathrm{avg}} = \frac{1}{T} \sum_{t_1+1}^{t_2} \gamma_i$ be the average learning rate between gradient steps $t_1$ and $t_2$. If $T$ is sufficiently large, that is

$$T \geq \frac{\log(1/\epsilon)}{\lambda \gamma_{\mathrm{avg}}} \tag{5}$$

then

$$w_{t_1}^{t_2} \leq \epsilon. \tag{6}$$

*Proof.* Without loss of generality, let $t_1 = 0$ and $T = t_2$. We have from equation (4):

$$w_0^T = \prod_{i=1}^{T}(1 - \gamma_i \lambda). $$

We want to guarantee that

$$\prod_{i=1}^{T}(1 - \lambda_i \gamma) \leq \epsilon. \tag{7}$$

First, apply the logarithm

$$\sum_{i=1}^{T} \log(1 - \gamma_i \lambda) \leq \log \epsilon. \tag{8}$$

Now, for $0 < x < 1$ it holds that $\log(1 - x) \leq -x$. This means that the inequality (7) is satisfied if

$$\lambda \sum_{i=1}^{T} \gamma_i \geq -\log \epsilon. \tag{9}$$

Denoting $\gamma_{\text{avg}} = \frac{1}{T}\sum_{i=1}^{T} \gamma_i$, this is the same as

$$T \geq \frac{\log(1/\epsilon)}{\lambda \gamma_{\text{avg}}}.$$

□

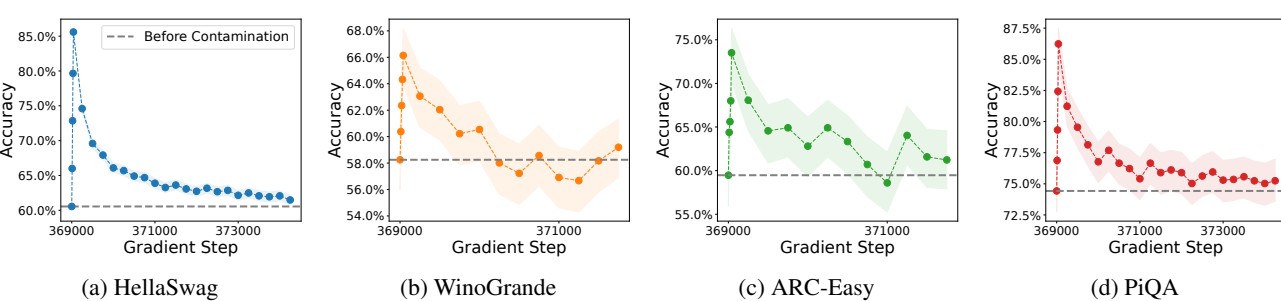

| (a) HellaSwag | (b) WinoGrande | (c) ARC-Easy | (d) PiQA |

*Figure 9.* **Contamination and forgetting in OLMo-1B.** We contaminate the OLMo-1B checkpoint at gradient step 369,000 four times with different benchmarks. This causes an average accuracy increase of 15 percentage points. We then continue pre-training for 1% of the remaining training time, leading to a reduction of 96% of the accuracy increase due to contamination. In this figure, different colours correspond to different benchmarks, and the grey line depicts the clean accuracy without contamination. Mean and bootstrapped 90% confidence intervals.

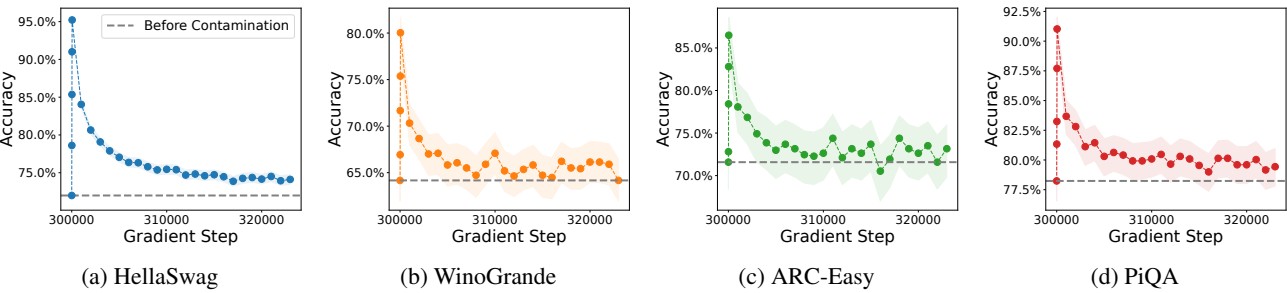

| (a) HellaSwag | (b) WinoGrande | (c) ARC-Easy | (d) PiQA |

*Figure 10.* **Contamination and forgetting in OLMo-7B.** We contaminate the OLMo-7B checkpoint at gradient step 300,000 four times with different benchmarks. This causes an average accuracy increase of 17 percentage points. We then continue pre-training for 13% of the remaining training time, leading to a reduction of 87% of the accuracy increase due to contamination. In this figure, different colours correspond to different benchmarks, and the grey line depicts the clean accuracy without contamination. Mean and bootstrapped 90% confidence intervals.

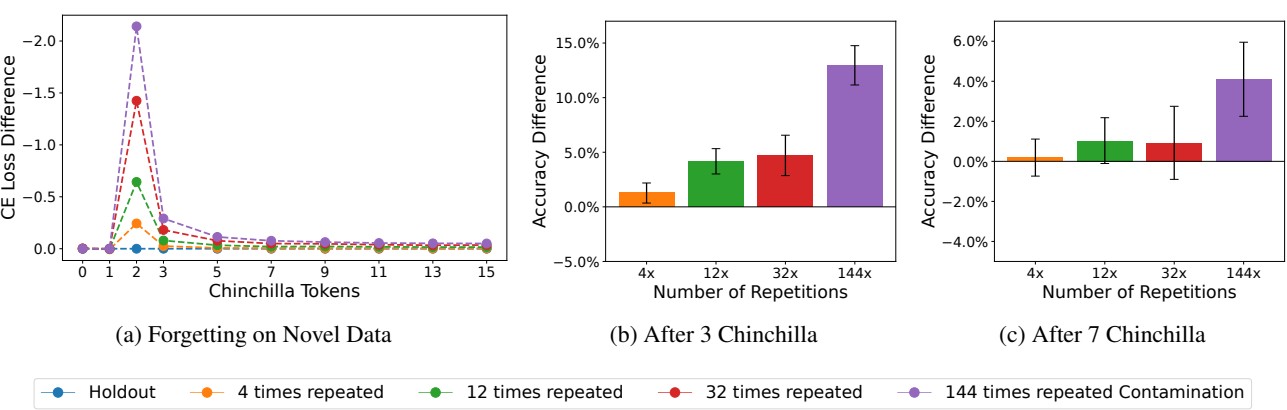

(a) Forgetting on Novel Data      (b) After 3 Chinchilla      (c) After 7 Chinchilla

*Figure 11.* **Forgetting without weight decay.** We perform the forgetting experiment from Section 4.2 without weight decay (instead of the default weight decay of 0.1). This figure depicts the same quantities as Figure 2 in the main paper. We see that there is significant forgetting without weight decay. At the same time, forgetting without weight decay occurs somewhat slower, especially for 144 times repeated contamination. In particular, the accuracy differences depicted in (b) are slightly larger than the corresponding accuracy differences in Figure 2, and the difference in (c) is statistically significant (compare Figure 2).

