# OpenReview forum: "How Much Can We Forget about Data Contamination?"
_ICML.cc/2025/Conference — ICML 2025 poster_

### Official Review · Reviewer_F1rX · 2025-03-10

**Overall Recommendation:** 4

**Summary:**

- the paper studies the effect of data contamination during the pre-training of language models, through a series of *controlled* contamination experiments
- the paper studies the effect by considering: (1) scaling the amount of contamination (i.e. repetitions), (2) scaling the model size, (3) scaling the (unrelated) pre-training data size, (4) scaling data and model jointly (Chinchilla), and (5) if and how weight decay plays a role
- the paper reports multiple findings, for example:
  - small-scale contamination may or may not be "forgotten" (i.e. the contamination stops contributing to benchmark gains), depending on how much (unrelated) pre-training data there are (as relative to Chinchilla optimal)
  - more repetitions of contamination uniformly increases benchmark performance, but the increase depends on both model scale and data scale
  - exposing to *novel* pre-training data is more effective at forgetting contamination than to old pre-training data
  - repetition is an important factor, perhaps arguably more so than seeing them later in training

## update after rebuttal

I appreciate the authors' rebuttal and will keep my score and my assessment that the paper should be accepted.

**Claims And Evidence:**

Yes, the claims in the paper are supported by extensive experiments to my understanding

**Essential References Not Discussed:**

N/A to my understanding. Key references are cited and discussed in the paper, although prior work's contributions can be highlighted a bit more.

**Experimental Designs Or Analyses:**

Overall, the experiments of the paper are well-designed to comprehensively answer the proposed research question. The findings are well supported by the experiment results.

Comments:
- (minor) deduplicating the contamination benchmark data (e.g. HellaSwag) from the pre-training tokens (e.g. FineWeb) is also recommended for a truly unbiased evaluation. Since the authors report accuracy gaps between holdout and contaminated, this is perhaps OK.
- (minor, mentioned earlier) depending on the model size (small vs big), pre-training data (FineWeb vs other pre-training mix), or even architecture (GPT-3 vs newer ones), the pre-trained model may not perform meaningfully on some of the benchmark data (e.g. the 124M model trained on 1x Chinchilla can be basically random guessing at MMLU). This could affect the takeaways, but on a macroscopic level the results are consistent and make sense.

**Methods And Evaluation Criteria:**

Yes, the proposed research question, methods, and evaluation data make sense overall.

- One minor weakness is that for some of the experiments, the authors used the 124M model, which is perhaps too small for meaningful performance on some of the benchmark datasets (e.g. MMLU).
- Another minor weakness is that FineWeb-Edu contains only text and largely no math or code; this means the models may be biased towards being a text model as opposed to a general purpose model like GPT-4. Though I understand that this could be an artifact of using LLM.c which primarily uses FineWeb-Edu.

**Other Comments Or Suggestions:**

- Section 4: "We being in" --> "We begin in"

**Other Strengths And Weaknesses:**

Strength
- the paper is well-written and a pleasure to read!

Weaknesses: see prior sections

**Questions For Authors:**

- Fig 1, 2: when you scale more pre-training data ("Chinchilla Tokens"), are the extra tokens *fresh* tokens, or repetitions of the same set of 1x Chinchilla tokens (e.g. same 20B tokens for an 1B model)? My understanding is fresh tokens, but since Fig 2 (b, c) says "Epochs", I'm not too sure

**Relation To Broader Scientific Literature:**

- this paper can be viewed as a more systematic, comprehensive, and rigorous execution of Jiang et al. (2024) [2] to understand the effect of data contamination in the pre-training stage
- related work like [1] discussed the effect of gradual forgetting of past training data over the course of training, which is related to this paper's finding that benchmark data can be forgotten (e.g. in abstract "Lllama 3 405B, have forgotten the data seen at the beginning of training.").
- this paper can be viewed as an intersection of several parts of the relevant literature: contamination analysis (many references in paper), memorization analysis (e.g. [1]), optimization analysis, data selection (e.g. use stale vs fresh pre-training data [3])


[1] https://arxiv.org/abs/2207.00099
[2] https://arxiv.org/abs/2401.06059
[3] https://arxiv.org/abs/2305.16264

**Theoretical Claims:**

The paper is mostly empirical. The analysis component for weight decay (sec 5.1) seems to make sense.

---

> ### Author Rebuttal · Authors · 2025-03-31
>
> Thank you for the detailed review of our paper. We are happy to hear that our paper is “a pleasure to read”! Below, we respond to your questions/comments.
>
> *“deduplicating the contamination benchmark data (e.g. HellaSwag) from the pre-training tokens (e.g. FineWeb) is also recommended for a truly unbiased evaluation.”*
>
> While it will not be possible to re-run the experiments with de-duplicated pre-training data, we commit to adding a Supplement Section that reports the overlap between the pre-training data and the benchmark questions that we contaminate with.
>
> Preliminary experiments with a random subset of benchmark questions and a random subset of the pre-training data indicate that the overlap between the benchmark questions and FineWeb-edu is small (as measured by the fuzzy string metric used to de-duplicate the benchmark questions).
>
>
>
> *“Key references are cited and discussed in the paper, although prior work's contributions can be highlighted a bit more.”*
>
> Thank you for the additional reference [3]; we will incorporate it. We think that the reviewer's positioning of our contribution in the literature is quite fitting! We will revise the relevant parts of the paper to highlight the contributions of prior work better.
>
>
>
> *“Fig 1, 2: when you scale more pre-training data ("Chinchilla Tokens"), are the extra tokens fresh tokens,”*
>
> Yes, they are fresh tokens. We will replace or qualify the usage of the term “epoch,” which can indeed be confusing in our setting.
>
>
> Thank you again for providing such a high-quality review. We would be happy to answer any additional questions.

---

> > ### Comment · Reviewer_F1rX · 2025-04-03
> >
> > I appreciate the authors' rebuttal and will keep my score and my assessment that the paper should be accepted.

---

### Official Review · Reviewer_T1JX · 2025-03-12

**Overall Recommendation:** 3

**Summary:**

The paper investigates the impact of data contamination in LLMs, specifically addressing whether small-scale contamination significantly affects benchmark evaluations. The authors analyze contamination effects along three dimensions: model size, number of training tokens, and repetition of contaminated examples. They find that while contamination can lead to overfitting under certain conditions, large-scale training beyond Chinchilla-optimal regimes can mitigate or even eliminate its impact. Empirical experiments demonstrate that continual pre-training can effectively erase contamination effects, with weight decay playing a key role in forgetting.

**Claims And Evidence:**

The experimental results support the paper’s main conclusions regarding the impact of data contamination and forgetting dynamics in large-scale training. However, the section on weight decay raises some questions, which will be discussed below.

**Essential References Not Discussed:**

The paper does not strongly rely on prior literature, so additional references are not essential.

**Experimental Designs Or Analyses:**

The experimental conclusions are sound and align well with expectations. No issues were found.

**Methods And Evaluation Criteria:**

The methods and evaluation criteria are reasonable for the problem. The main limitation is the relatively small scale of experiments, though this is understandable given computational constraints.

**Other Comments Or Suggestions:**

First line of Section 4: "We being" -> "We begin"

**Other Strengths And Weaknesses:**

**Strengths:**
1. The paper includes a large number of experiments with well-designed and solid methodologies, leading to conclusions that align with expectations.
2. This paper is well-written.

**Weaknesses:**
1. The practical actionable insights of these empirical findings are unclear.
2. The experiments are limited in scale (model size), though this is understandable given computational constraints.
3. The analysis of weight decay is relatively shallow. While the perspective is interesting, the analysis mainly focuses on gradient decay, ignoring how earlier gradients influence the optimization trajectory. Simply showing that gradients decay to zero may not be sufficient. Additionally, the paper states that weight decay is not necessary for forgetting, making the core insights of Section 5 unclear. Does this section propose a possible explanation? If so, it is presented in a shallow way and is not clearly necessary, raising questions about its contribution.

Another point of confusion is the mention of data attribution. While the paper references it, neither the experimental results nor the theoretical analysis provide any direct connection to data attribution methods. This is not necessarily a weakness, but it raises questions about why data attribution is discussed if it does not play a role in the findings.

**Questions For Authors:**

1. Your experiments provide insights into LLM forgetting, but it is unclear what **actionable** takeaways arise from these findings. How do you envision these results informing model training or evaluation practices?
2. Section 5 suggests weight decay contributes to forgetting, but the analysis is mainly based on gradient decay without considering how earlier gradients influence the optimization trajectory. Given that the paper also states weight decay is not necessary for forgetting, what is the key insight of this section? Are you proposing weight decay as a primary mechanism, or just one possible factor?
3. The paper references data attribution, but the experiments and theoretical analysis do not directly connect to it. What role does data attribution play in your findings, and how does it relate to the core contributions of the paper?

Overall, I really appreciate the experimental design of this paper. If Questions 1 and 2 are addressed convincingly, I would be happy to increase my score.

**Relation To Broader Scientific Literature:**

The paper provides findings on LLM forgetting through controlled experiments, contributing to the understanding of data contamination and memory dynamics. However, it is unclear how these conclusions translate into actionable insights, which I would like to ask the authors about.

**Theoretical Claims:**

I checked Proposition 1, and it appears to be correct. No issues were found.

---

> ### Author Rebuttal · Authors · 2025-03-31
>
> Thank you for the detailed review of our paper and insightful questions. Below, we give detailed answers to your questions/comments.
>
> *“The main limitation is the relatively small scale of experiments, though this is understandable given computational constraints.”*
>
> Running experiments with a 7B parameter model is the largest we can afford on an academic compute budget. We hope that, inspired by our work, future work will study forgetting and contamination as part of a large-scale training run.
>
> *“Your experiments provide insights into LLM forgetting, but it is unclear what actionable takeaways arise from these findings. How do you envision these results informing model training or evaluation practices?”*
>
> A key insight of our work is that the causal effect of a small number of data points in the LLM pre-training data on final model behavior can be zero. If model training is in this large-data regime, this has several important practical implications:
>
> - Our work questions the common practice of grouping benchmark questions into “clean” and “contaminated” questions based on the maximum found overlap between a question and the training data (for example, n-gram overlap) to reason about benchmark overfitting (Brown et al. 2020, Touvron et al., 2023, Dubey et al., 2024). Specifically, our work shows that the mere existence of an evaluation data point in the LLM pre-training data is not necessarily an indicator of benchmark overfitting. We show that it is necessary to consider other factors, such as the frequency of the contamination.
> - Our work highlights the importance of when a contaminated sample is seen. If the samples are seen early during pre-training, overfitting is mitigated due to the forgetting phenomenon. This is directly **actionable**: If we suspect benchmark overfitting during an early pre-training stage on relatively unfiltered internet data, the practitioner can consider continual pre-training (“mid-training”) on clean data to mitigate the overfitting.
> - Our work suggests that filtering benchmark questions from the mid-training and post-training data is much more critical than filtering the pre-training data (additional factors like the learning rate schedule come into play). This has immediate and direct implications for the design of training datasets.
>
> We would be happy to further elaborate on these points.
>
>
> *“Section 5 suggests weight decay contributes to forgetting, but the analysis is mainly based on gradient decay without considering how earlier gradients influence the optimization trajectory. [...] what is the key insight of this section? Are you proposing weight decay as a primary mechanism, or just one possible factor?”*
>
> Let us outline the motivation for Section 5 in the paper and our interpretation of the experimental results.
>
> Given that forgetting is a key empirical property of LLM pre-training, it is natural to ask: Is there a simple factor in the pre-training pipeline that explains the forgetting? Asking this question, it seemed straighforward to consider the weight decay parameter: After all, weight decay is a mechanistic process that gradually removes the influence of earlier gradient updates on the model weights (this is what we formally describe in Proposition 1).
>
> In our experiments, it turned out that the weight decay parameter does indeed influence forgetting in the sense that increasing the weight decay parameter leads to faster forgetting (Figure 4). The experiments also revealed that the empirical rate of forgetting occurs faster than the cumulative weight decay. This makes a lot of sense to us, given how weight decay works mechanistically. At the same time, the experiments also demonstrated that weight decay is not necessary for forgetting (Supplement Figure 8) – though forgetting without weight decay occurs at a slower rate. We don’t see this as a problem for our analysis—it simply means that forgetting is a multifaceted phenomenon not exclusively driven by weight decay.
>
> Section 5.3 suggests that the mechanism of forgetting via weight decay is likely relevant in large-scale LLM pre-training.
>
> To summarize, the key insight in Section 5 is that the weight decay parameter has a causal effect on forgetting that is likely relevant in large-scale LLM pre-training.
>
> *“What role does data attribution play in your findings, and how does it relate to the core contributions of the paper?”*
>
> Our paper's controlled experiments resemble the “leave-one-out” (LOO) procedure in data attribution. Concretely, our experiments directly “leave-out” entire groups of benchmark questions, thus simulating a kind of “group” LOO procedure.
>
> Moreover, our result that the (average) causal effect of inserting individual benchmark questions into the pre-training data can be zero suggests that we might not be able to attribute “importance” to individual data points in the LLM pre-training regime.
>
> We would be happy to answer any additional questions.

---

> > ### Comment · Reviewer_T1JX · 2025-04-07
> >
> > Thank you for the detailed and thoughtful response. I’d like to follow up on a few points:
> >
> > **Regarding Q1**: I think the points you raised—such as the effect of contamination frequency—are certainly valid, but also somewhat expected. I’m curious whether your findings offer any implications for the design of **benchmark contamination detection** methods. For instance, could frequency- or position-aware strategies improve beyond simple overlap-based heuristics? I understand this may be future work, but hearing your thoughts would strengthen the **practical** relevance of your conclusions.
> >
> > As for the role of when a contaminated sample appears in training: while the result aligns with intuition, I understand this is exactly the kind of effect that controlled experiments like yours can rigorously isolate and quantify. I do see value in that.
> >
> > **Regarding Q2**: I now better understand the intention behind Section 5 and appreciate the mechanistic perspective on weight decay and forgetting. However, as a standalone section, I still feel the analysis is relatively shallow—the theoretical side does not go very deep, and the practical takeaway remains somewhat limited. I wonder if this section could benefit from a clearer articulation of how one might use these findings to **predict or control** forgetting in large-scale training.
> >
> > To be clear, I am not nitpicking—I genuinely like this paper a lot. The experimental design and metrics are elegant and thoughtfully constructed, and I find the section on weight decay particularly interesting. It’s precisely because the paper covers so much ground that I find myself thinking harder about what the key **practical takeaways** might be. In fact, because the content is quite rich, I would suggest explicitly summarizing your takeaways—perhaps in a dedicated section, even in the appendix—so that readers can more easily understand the implications for practice. I want to emphasize that I recommend acceptance; my comments are offered in the spirit of exploring how to make an already strong paper even more impactful.
> >
> > One additional limitation, which is perhaps unavoidable for this type of controlled study, is that real-world LLM training involves a wide range of model architectures, scales, and benchmark types (e.g., question formats, domains). While your experiments already cover multiple models and datasets, they inevitably cannot represent the full diversity of practical settings. As a result, findings such as “the causal effect of a single data point can be zero” may not universally hold. I would suggest adding a short discussion of this limitation to clarify the boundaries of the conclusions and avoid overgeneralization.
> >
> > I have increased my score to a 3. While some aspects could be further clarified or expanded (e.g., practical takeaways and the weight decay analysis), I see these as opportunities for refinement rather than fundamental flaws. I welcome any further discussions.

---

> > > ### Author Response · Authors · 2025-04-08
> > >
> > > Thank you for the thoughtful comment and for increasing your score.
> > >
> > > To us, the most important takeaway from the paper is the negative result: If the scale of the data is large in comparison to the scale of the model, contamination can be completely forgotten.
> > >
> > > Now, it is true that this insight does not boil down to a single recommendation of the form “do X” (as in, “add qk-norm to increase training stability”). However, we would argue that it is highly relevant to several practices in pre-training (as outlined in our original rebuttal comment) and also to how we think about LLMs more generally.
> > >
> > > One example is contamination detection methods, where our paper primarily provides arguments to be critical of these methods. Consider, for example, the widely used n-gram overlap. In our experiments, if we were to compute the n-gram overlap between the pre-training data and the benchmark questions used for evaluation, we would find a large n-gram overlap for the questions we contaminate with (by construction) and a much smaller overlap for the holdout questions (see our response to Reviewer F1rX). Now, at least in the setting that we study in our paper, the n-gram overlap does not determine overfitting. Instead, what is relevant is the joint scale of model, data, and contamination. By modifying the scale of the model, the data, and the number of repetitions of benchmark questions, we can get any possible result from no overfitting to complete overfitting while the n-gram overlap between the pre-training data and the contaminated benchmark questions is consistently large.
> > >
> > > Concerning Section 5, we completely agree with the reviewer that more research is needed on how cumulative weight decay can predict forgetting in large-scale training. We decided to include Section 5 in the paper, even if it is relatively brief, since it provides a valuable backdrop to the experiments in the previous Sections.
> > >
> > > Perhaps one final observation: When we write, “We hope that, inspired by our work, future work will study forgetting and contamination as part of a large-scale training run.” we actually think it would be important to do this. Among others, this is because of the point raised by the reviewer, namley it would be desirable to corroborate our results with evidence from “real-world” LLM training.
> > >
> > > *“I would suggest explicitly summarizing your takeaways—perhaps in a dedicated section, even in the appendix  [...]  I would suggest adding a short discussion of this limitation to clarify the boundaries of the conclusions and avoid overgeneralization.”*
> > >
> > > This is a great idea. We commit to adding two new sections to the appendix, one to discuss the takeaways and one to discuss the limitations of our work.
> > >
> > > Thank you again for the interesting comment! We appreciate that this kind of discussion helps us to clarify the contribution of the paper and and make it more impactful.

---

### Official Review · Reviewer_Mm41 · 2025-03-12

**Overall Recommendation:** 4

**Summary:**

This paper investigates the impact of data contamination in large language models (LLMs), challenging the assumption that minor contamination invalidates benchmark evaluations. Through controlled experiments, the authors study how contamination effects scale with model size (up to 1.6B parameters), training tokens (up to 40B), and example repetitions (up to 144x).

**Claims And Evidence:**

The claims are supported by systematic experiments and theoretical analysis. Evidence includes:

1. Controlled scaling experiments showing monotonic trends in contamination effects.

2. Weight decay analysis linking optimization hyperparameters to forgetting.

3. Validation via OLMo-7B continual training However, extrapolation to larger models (e.g., Llama 3 405B) relies on theoretical weight decay bounds rather than direct empirical validation.

**Essential References Not Discussed:**

None

**Experimental Designs Or Analyses:**

Experiments are methodical but limited to smaller models due to computational constraints. The extrapolation to larger models is plausible but unverified empirically. The OLMo-7B experiments add credibility, but testing on >10B parameter models would strengthen conclusions.

**Methods And Evaluation Criteria:**

Benchmarks (ARC-Easy, HellaSwag, etc.) are filtered for duplicates to isolate contamination effects. Contamination is inserted randomly, mimicking real-world leakage. Evaluation via accuracy gaps between contaminated and holdout data is appropriate. A limitation is the focus on smaller models (≤7B), which may not fully capture dynamics in larger LLMs.

**Other Comments Or Suggestions:**

I recommend that the authors open-source the code and experimental data logging of this work in order to verify reproducibility and increase the transparency of the work.

**Other Strengths And Weaknesses:**

None

**Questions For Authors:**

Could paraphrased or semantically similar contamination (vs. exact matches) alter the conclusions, especially for larger models?

For example the "Semantic Level" and "Information  Level" contamination in the Xu et al. 2024 (Benchmark data contamination of large language models: A survey)

**Relation To Broader Scientific Literature:**

None

**Theoretical Claims:**

Proposition 1 (cumulative weight decay bounds forgetting) is proven in the appendix. The proof assumes constant learning rates, which may not hold in practice, but the core intuition—weight decay reduces past gradient influence—is valid.

---

> ### Author Rebuttal · Authors · 2025-03-31
>
> Thank you for the detailed review of our paper. We are happy to hear that you appreciate our experimental design. Below, we give answers to your questions/comments.
>
> *“The proof assumes constant learning rates”*
>
> To clarify, the proof does not assume constant learning rates (in the proof, the learning rate is denoted $\gamma_i$ and depends on the gradient step). We conjecture that the reviewer observed that $\lambda$ in the proof is constant - this is the weight decay (which is usually constant, but the algebra in the proof does not require this).
>
> *“I recommend that the authors open-source the code and experimental data logging of this work in order to verify reproducibility and increase the transparency of the work.”*
>
> The code is available at https://github.com/icml9771/code (the link is currently in the supplement; we will move it to the first page). We additionally commit to open-source our Weights & Biases logs and model checkpoints.
>
> *“Could paraphrased or semantically similar contamination (vs. exact matches) alter the conclusions, especially for larger models?”*
>
> This is an interesting question. The reviewer is correct in observing that we decided to consider only exact contamination because non-exact contamination might behave qualitatively differently for larger models (see also Supplement A.1. Additional Discussion of Data Contamination Assumptions and Setting). For example, larger models might observe a kind of ``emergence’’ phenomenon where they can suddenly make efficient use of rephrased samples.
>
> That being said, we agree with the provided reference Xu et al. (2024), which states that exact contamination is more severe than other forms of contamination (e.g. the last paragraph on page 4 in Xu et al. (2024)). This means that the results in our paper should provide a heuristic upper bound for what would happen with paraphrased or semantically similar contamination.
>
> Of course, ultimately experimental evidence would be required for other forms of contamination, too.
>
> We would be happy to answer any additional questions.

---

> > ### Comment · Reviewer_Mm41 · 2025-04-01
> >
> > Thanks for the author's response, which addressed my concerns.

---

### Official Review · Reviewer_KE3e · 2025-03-14

**Overall Recommendation:** 4

**Summary:**

This paper provides a very important perspective in data contamination of LLM and show that not all data leakage will lead to false evaluation in benchmarks.

**Claims And Evidence:**

Strengths:
1. This paper question the severity mentioned in the previous paper. The assumption or the settings of data contamination might not be practical or not common in LLM.
2. Besides the problem, this paper gives a comprehensive evaluation on the property of forgetting.
3. The paper is well-written and the problem is interesting.

Question:
1. Although the paper point out this problem, people still have the question when would the benchmark be totally safe. And is there any method to improve or gaurantee the fairness of the benchmark?

**Essential References Not Discussed:**

N/A

**Experimental Designs Or Analyses:**

The experiments are comprehensive.

**Methods And Evaluation Criteria:**

Yes. Their evaluation is more practical than the previous paper.

**Other Comments Or Suggestions:**

N/A

**Other Strengths And Weaknesses:**

See above

**Questions For Authors:**

N/A

**Relation To Broader Scientific Literature:**

It questions the reaonability in previous settings of data containation paper.

**Theoretical Claims:**

N/A

---

> ### Author Rebuttal · Authors · 2025-03-31
>
> Thank you for reviewing our paper and for your positive assessment.
>
> It seems that you ask under what conditions we can be confident that a benchmark evaluation is not contaminated. This is an interesting question that lies somewhat beyond the scope of our paper. In our paper, we demonstrate that moderate amounts of exact contamination do not necessarily lead to benchmark overfitting. This implies that minor mistakes in data pre-processing and filtering might not lead to benchmark overfitting. To provide stronger guarantees that a benchmark evaluation is "totally safe", we would require a much deeper understanding of the learning dynamics of LLMs.
>
> We would be happy to answer any additional questions.

---

### Decision · Program_Chairs · 2025-05-01

**Decision:**

Accept (poster)

**Comment:**

This paper investigates the impact of data contamination in large language models (LLMs), challenging the assumption that minor contamination invalidates benchmark evaluations. Through controlled experiments, the authors study how contamination effects scale with model size (up to 1.6B parameters), training tokens (up to 40B), and example repetitions (up to 144x).

The paper is well-written and includes a large number of experiments with well-designed and solid methodologies. The weight decay analysis linking optimization hyperparameters to forgetting is interesting. Validation is conducted via OLMo-7B continual training. The experiments are limited in scale (model size), though this is understandable given computational constraints.

The paper reports multiple findings:
(i) small-scale contamination may or may not be "forgotten" (i.e. the contamination stops contributing to benchmark gains), depending on how much (unrelated) pre-training data there are (as relative to Chinchilla optimal)
(ii) more repetitions of contamination uniformly increases benchmark performance, but the increase depends on both model scale and data scale
(iii) exposing to novel pre-training data is more effective at forgetting contamination than to old pre-training data
(iv) repetition is an important factor, perhaps arguably more so than seeing them later in training

Recent, very related work:
[1] Overestimation in LLM Evaluation: A Controlled Large-Scale Study on Data Contamination’s Impact on Machine Translation
https://arxiv.org/pdf/2501.18771?